

# Analysis of sea ice deformation and influencing factors in the western Arctic from 2022 to 2023

Xiaomin Chang[1], Lei Ji[1], Bowen Zhu[1], Guangyu Zuo[2], Yinke Dou[2], Tongliang Yan[1]

[1]College of Water Conservancy Science and Engineering, Taiyuan University of Technology, No. 79 Yingze West Street, Taiyuan 030024, China

[2]College of Electrical and Power Engineering, Taiyuan University of Technology, No. 79 Yingze West Street, Taiyuan 030024, China

*Correspondence to*: Xiaomin Chang (changxiaomin@tyut.edu.cn)

**Abstract.** Sea ice governs the global climate, safeguards ecological equilibrium in polar regions, and influences ocean circulation; however, the factors impacting the spatial characteristics of sea ice deformation have not been comprehensively analyzed. This study examined the effects of wind speed, air temperature (T2m), and sea ice thickness on the variation in sea ice deformation within the western Arctic based on the Lagrangian diffusion theory using buoy data from March 2022 to March 2023. The total sea ice deformation gradually declined in all seasons except for the melting season, especially in the fall and winter. Owing to the ongoing sea ice consolidation, the average total deformation was lower in the fall and winter than in the spring. The total deformation of sea ice diminished as the spatial scale increased. As sea ice is thinner on average in spring, geostrophic winds are the primary factor influencing the spatial characteristics of sea ice deformation. In contrast, the larger average ice thickness in fall and winter reduces the significance of the external force, and T2m/sea ice thickness is the primary factor influencing the spatial characteristics of sea ice deformation. Our multivariate nonlinear regression model effectively predicted the total sea ice deformation. This study provides a scientific basis for climate change research, sea ice change prediction, climate model validation, resource management, and environmental protection.

## 1 Introduction

Arctic sea ice is critical for the local ecosystem, with significant implications for global climate, ocean circulation, resource development, and international cooperation (Vihma, 2014; Brown et al., 2017; Ceppi et al., 2017). In recent years, the ice extent and average ice thickness of the Arctic sea have significantly decreased (Lindsay et al., 2015; Kwok et al., 2015). Notably, the weakening of internal ice stresses has increased the drift speed of ice floes, although winds do not exhibit significant changes (Rampal et al., 2009; Spreen et al., 2011). Sea ice drift driven by winds and currents is heterogeneous and leads to ice breakup and deformation. Sea ice typically deforms and redistributes during atmospherically driven events such as storms or sudden changes in wind direction (Hutchings et al., 2011; Itkin et al., 2017).

As the strength of the sea ice interior weakens and the drift speed increases, more sea ice is deformed, which positively affects sea ice mass balance (Herman et al., 2012). Previous studies have analyzed sea ice dynamics. For example, Stern et al. (2009)



reported that the mean deformation rate of Arctic sea ice is inversely associated with the spatial scale, following a power law with an exponent of approximately -0.2. Notably, seasonal and interannual variations in the spatial scale index β, which is used to characterize the spatial features of sea ice deformation, have been reported (Stern et al., 2009; Herman et al., 2012). The rate of sea ice deformation, its spatiotemporal scaling index, and the localization of ice deformation decrease with the freezing

of ice floes (Herman et al., 2012). This exponent becomes more negative during the summer, with the deformation rate reaching its minimum in the following winter. Rampal et al. (2008) further elaborated that Arctic sea ice deformation follows specific space- and time-scaling laws, and the deformation rate depends on observation scales. They also highlighted time–space coupling, where the time-scaling exponent varied with the spatial scale and vice versa. In addition, Hutter et al. reported that simulated sea ice deformation exhibits multiple fractal spatial scales consistent with satellite data, suggesting that the

complexities of the spatial characteristics of deformation vary at different scales (Mohammadi-Aragh et al., 2020).

Multiple equilibrium flow states may exist in the Arctic Basin, the characterization of which is influenced by sea–ice strength and ice rheology (Hibler et al., 2006). Lei et al. (2020) reported a higher ice deformation rate in 2014/2015 and 2016/2017 than that observed during the Surface Heat Budget of the Arctic Ocean (SHEBA) programme in 1997/1998, which was attributed to recent summer ice loss, particularly thick multi-year ice (Herman et al., 2012). Wang et al. (2021) highlighted

wind forcing as a significant influencer of sea ice characteristics, including drift, thickness, concentration, and deformation rates. Moreover, the spatial scale pattern of ice deformation during summer depends on ice conditions and the wind and spatial homogeneity of ice motion (Lei et al., 2020). In addition, the wind–ice deformation correlation was highest at the basin scale and decreased with the size of the study area (Herman et al., 2012). Sea ice deformation is an important process in the Arctic climate system. The formation of openings and pressure ridges in sea ice in response to external forces drives changes in the

distribution of ice thickness, thereby controlling the exchange of heat, moisture, and momentum between the ocean and the atmosphere. Enhanced winter sea ice deformation may increase ice production (Stern et al., 1995; Thorndike et al., 1975) as well as momentum and energy exchange between the atmosphere and ocean by increasing the intermittent opening of the ice sheet (Heil et al., 2002). Therefore, understanding the spatial and temporal characteristics of sea ice deformation and monitoring the associated influencing factors are crucial for predicting future changes in the Arctic ice sheet and addressing

global climate change.

Sea ice deformation rates can often be determined by tracking objects on ice, such as GPS-located buoys or natural objects identified using radar imagery (Oikkonen et al., 2016; Hutchings et al., 2012). However, owing to the low spatial coverage of radar imagery, the tracked objects mainly provide information on small-scale ice deformation (Oikkonen et al., 2016; Oikkonen et al., 2017). Notably, studies on the factors affecting sea ice deformation characteristics at large ranges of spatial and temporal

scales are limited. In addition, although sea ice models can successfully reproduce the observed mean ice motion, they exhibit limitations in terms of ice deformation (Herman et al., 2012).

The total sea ice deformation rate obtained using numerical models is approximately 40% lower than that from satellite observations, particularly in seasonal sea ice areas (Spreen et al., 2017). Therefore, to compensate for the limitations of radar





images in monitoring sea ice deformation on a vast scale, meticulous examination of the factors influencing the spatial

characteristics of Arctic sea ice deformation across different spatial scales and seasons is imperative.

This study aimed to analyse the sequence of changes in sea ice deformation in the western Arctic during March 2022–March 2023 using ice-based buoy data based on the Lagrangian diffusion theory and highlight the spatial and temporal variations in atmospheric conditions in the study area. Furthermore, this study used the spatial scaling index β to characterize the spatial features of sea ice deformation and analyzed the factors influencing the spatial features of sea ice deformation at two different

sets of spatial scales (5–100 and 130–400 km) using remote-sensing data, which have important guiding roles in the study of climate change, ecosystem management, and environmental protection. In addition, owing to the limitations of the sea ice model in predicting ice deformation, this study constructs a preliminary multiple regression model to predict the rate of sea ice deformation to compensate for the difficulties in practical calculations.

## 2 Data and methods

### 2.1 Data

### 2.1.1 Buoy data

All buoy observations in this study were from the International Arctic Buoy Programme (IABP) aimed at maintaining a network of drifting buoys in the Arctic Ocean to provide meteorological and oceanographic data for real-time operational requirements and research purposes, including support for the World Climate Research Program and the World Weather Watch

Program. The buoy types used in this study are listed in Table 1. The sampling frequency of different buoys varies from 0.5 to 1 h, and a GPS positioning system is usually used with a positioning error of <10 m.



**Table 1: Details of the buoys selected for the study**

| Buoy ID | Buoy Type | Organization | Starting time | Starting longitude (°E) | Starting latitude (°N) | Ending Time | Ending Longitude (°E) | Ending latitude (°N) |
|---|---|---|---|---|---|---|---|---|
| 1 | DOT | PSC | 2022/3/17 | -144.50 | 72.54 | 2022/5/10 | -156.10 | 73.07 |
| 2 | SK | Jamstec | 2022/3/17 | -144.49 | 72.54 | 2022/8/14 | -151.54 | 76.52 |
| 3 | IT | Jamstec | 2022/3/17 | -144.31 | 72.47 | 2022/10/23 | -165.46 | 78.87 |
| 4 | IT | Jamstec | 2022/3/17 | -144.49 | 72.63 | 2022/11/8 | -152.44 | 80.86 |
| 5 | IT | Jamstec | 2022/3/17 | -144.20 | 72.57 | 2023/2/16 | -152.38 | 81.43 |
| 6 | UT | Jamstec | 2022/3/17 | -144.49 | 72.54 | 2022/9/17 | -155.06 | 79.62 |
| 7 | ITB | USIABP | 2022/4/3 | -155.69 | 72.53 | 2022/7/12 | -162.76 | 74.51 |
| 8 | Ice Ball | USIABP | 2022/4/3 | -154.27 | 81.40 | 2022/11/7 | -166.78 | 79.45 |
| 9 | Ice Ball | USIABP | 2022/6/29 | -149.66 | 74.99 | 2022/10/16 | -154.87 | 79.07 |
| 10 | IP | USIABP | 2022/4/6 | -156.40 | 71.97 | 2022/9/23 | -171.06 | 77.59 |
| 11 | AXIB_53 | USIABP | 2022/10/12 | -150.01 | 73.05 | 2023/3/24 | -173.15 | 79.39 |
| 12 | Ice Ball_33 | USIABP | 2022/3/17 | -142.73 | 71.64 | 2023/3/24 | 174.11 | 80.58 |
| 13 | Ice Ball_34 | USIABP | 2022/3/17 | -142.96 | 71.99 | 2022/9/4 | -157.64 | 77.06 |
| 14 | Ice Ball_37 | USIABP | 2022/3/17 | -145.10 | 72.07 | 2023/2/19 | 176.95 | 79.44 |
| 15 | IT | USIABP | 2022/10/12 | -149.77 | 76.07 | 2023/3/24 | -158.05 | 80.56 |
| 16 | IT | USIABP | 2022/10/12 | -149.80 | 75.67 | 2023/3/24 | -159.19 | 80.58 |
| 17 | IT | USIABP | 2022/10/12 | -149.85 | 75.13 | 2023/3/24 | -162.08 | 80.40 |
| 18 | IT | USIABP | 2022/10/12 | -149.87 | 74.73 | 2023/3/24 | -165.82 | 80.18 |



### 2.1.2 Reanalysis of data

This study used the European Centre for Medium-Range Weather Forecast (ECMWF) Reanalysis v5 (ERA5) atmospheric reanalysis data (http://dx.doi.org/10.24381/cds.adbb2d47). We analyzed the atmospheric forcing characteristics at the basin scale and along the buoy. This dataset provided reanalysis data since 1979 and is the fifth generation of atmospheric reanalysis data released by the ECMWF. The horizontal resolution was 0.25°, the vertical layering was 137 layers, and the vertical resolution increased to 0.01 hPa (approximately 80 km), with a temporal resolution of 1 h. The reanalysis data used in this study included 10-m wind field data (10 m wind), sea level pressure (SLP) data, and 2 m air temperature data from 1979 to 2023. The hourly wind speed, T2m, and sea-level air pressure were calculated by interpolating the reanalyzed data to match the buoy position.

To clarify the sea ice conditions around each buoy at different times, the sea ice concentration data used in this study were sourced from the National Snow and Ice Data Center (NSIDC), and the dataset selected was the long-term Bootstrap Sea Ice Concentrations from Nimbus-7 Scanning Multichannel Microwave Radiometer (SMMR) and Defense Meteorological Satellite Program (DMSP) Special Sensor Microwave/Imager (SSM/I)-Special Sensor Microwave Imager/Sounder (SSMIS) v4. The spatial resolution of this dataset was 25 km, with a temporal resolution of 1 day, spanning from November 1978 to March 2023. This dataset was derived from measurements obtained using the SMMR on the Nimbus-7 satellite and the SSM/I sensor on the DMSP satellites F8, F11, and F13. It also includes measurements from the SSMIS on the DMSP-F17. The sea ice concentration in this dataset was derived from a modified Bootstrap algorithm that used a set of dynamically adjusted sea ice and open-water tie points. Comiso et al. (2017) analyzed two passive microwave algorithms for sea ice concentration: the Bootstrap Algorithm and the NASA Team Algorithm. For more information on algorithm, please refer to their studies on passive microwave algorithms for sea ice concentration (Steele et al., 2015).

The sea ice thickness dataset used in this study was the SMOS-CryoSat L4 Sea Ice Thickness v206, released by the European Space Agency in 2009. The SMOS-CryoSat merges the Sea Ice Thickness Level 4 product based on estimates from both MIRAS and SIRAL instruments, with a significant reduction in the relative uncertainty for the thickness of the thin ice. All grids were projected onto a 25 km EASE2 grid based on a polar aspect spherical Lambert azimuthal equal-area projection. The grid dimensions were 5,400 × 5,400 km, equivalent to a 432 × 432 grid centred on the geographic pole.

### 2.2 Methods

Three methods were developed to estimate the ice strain rate from the data collected by ice drifters. Two methods use the Lagrangian statistics of single-particle absolute dispersion (Lukovich et al., 2011) or the relative dispersion of buoy pairs (Rampal et al., 2008). The third method computes the differential kinematic properties (DKPs) from data collected using denser buoy clusters (Hutchings et al., 2008; Heil et al., 2007) and provides synchronous estimates of the normal and shear components that cannot be obtained from calculations based on single- or double-particle dispersions (Lukovich et al., 2017).





Lagrangian diffusion theory is widely applied in the fields of atmospheric and ocean dynamics to describe the topology and dynamic characteristics of flow fields. In the study of sea ice motion, the Lagrangian diffusion theory is applied to quantitatively depict the motion and deformation processes of sea ice by analyzing data from ice-based buoys (Colony et al., 1984; Colony et al., 1985; Lukovich et al., 2011; Lukovich et al., 2017; Rampal et al., 2016; Rampal et al., 2009; Rampal et al., 2008). Based on the number of particles studied, diffusion can be further divided into single- (absolute), two- (relative), and three-particle diffusion.

These three research methods can be used for sea ice deformation analysis in different scenarios, as required. This study primarily adopted the third diffusion method—three-particle diffusion—to calculate sea ice deformation parameters in the Arctic region and investigate the deformation characteristics of sea ice in depth. The deformation rate of the triangular region formed by the three buoys was calculated using the three-particle method. The differential motion characteristics of the entire triangular region were obtained by computing the movement of the three vertices of the triangle formed by the buoy array in the x- and y-directions using Green's theorem to calculate the line integrals. Compared with the first two methods, the buoy triangle array can simultaneously provide more detailed information on sea ice divergence and shear characteristics.

In this study, we utilized each buoy in the study area to arbitrarily construct a triangular network and used arbitrary triangles as the basic unit of deformation calculation. The sea–ice deformation was calculated using the sea–ice drift velocity at the vertices of the triangles ($u_i, v_i$ ($i = 1, 2, 3$)), as shown in Figure 1b.

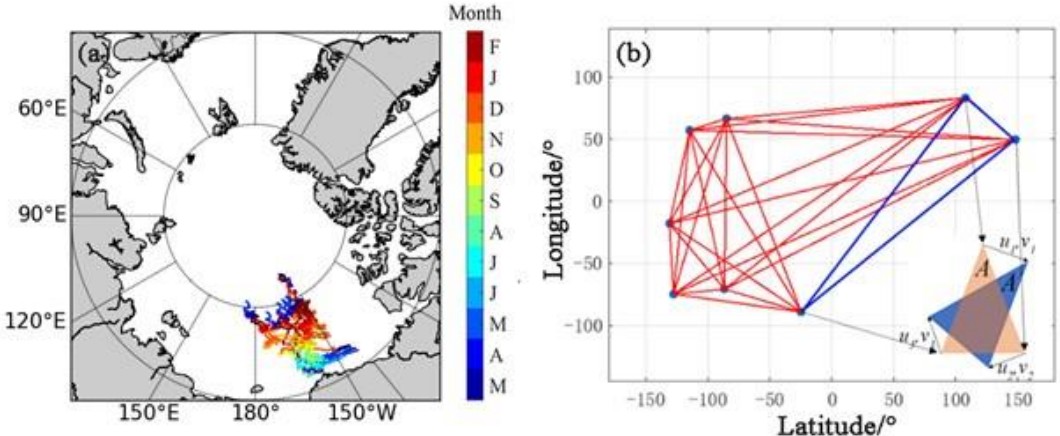

**Figure 1: Schematic diagram of buoy triangular mesh and the basic triangular unit for deformation calculation.**

To ensure that the triangulation network obtained the correct sea ice motion characteristics and reduce calculation errors, this study screened the triangles involved in the deformation calculation. The screening conditions were as follows: (1) the interior angle of the triangle is >15°. (2) The sea ice drift speed at the vertices was >0.02 m/s. (3) The order of the three vertices of the



triangle satisfies the right-hand rule (counter clockwise). Subsequently, for a triangular area A, the deformation rate

components were obtained using the divergence theorem, as follows:

$$\begin{cases} \frac{\partial u}{\partial x} = \frac{1}{A} \oint u\, dy \\ \frac{\partial u}{\partial y} = -\frac{1}{A} \oint u\, dx \end{cases} \tag{1}$$

$$\begin{cases} \frac{\partial v}{\partial x} = \frac{1}{A} \oint v\, dy \\ \frac{\partial v}{\partial y} = -\frac{1}{A} \oint v\, dx \end{cases} \tag{2}$$

Eqs. (1) and (2) can be approximated as follows:

$$\begin{cases} \frac{\partial u}{\partial x} = \frac{1}{2A}[(u_1 + u_3)(y_1 - y_3) + (u_1 + u_2)(y_2 - y_1) + (u_2 + u_3)(y_3 - y_2)] \\ \frac{\partial u}{\partial y} = -\frac{1}{2A}[(u_1 + u_3)(x_1 - x_3) + (u_1 + u_2)(x_2 - x_1) + (u_2 + u_3)(x_3 - x_2)] \end{cases} \tag{3}$$

$$\begin{cases} \frac{\partial v}{\partial x} = \frac{1}{2A}[(v_1 + v_3)(y_1 - y_3) + (v_1 + v_2)(y_2 - y_1) + (v_2 + v_3)(y_3 - y_2)] \\ \frac{\partial v}{\partial y} = -\frac{1}{2A}[(v_1 + v_3)(x_1 - x_3) + (v_1 + v_2)(x_2 - x_1) + (v_2 + v_3)(x_3 - x_2)] \end{cases} \tag{4}$$


The area A of the triangle was calculated using Eq. (5), as follows:

$$A = \frac{1}{2}[(x_1 y_2 - x_2 y_1) + (x_2 y_3 - x_3 y_2) + (x_3 y_1 - x_1 y_3)] \tag{5}$$

Using Eqs. (3) and (4), the deformation rate parameters—divergence (div) and shear (shr)—were calculated with their following computational formulas in Eqs. (6) and (7), respectively:


$$div = \frac{\partial u}{\partial x} + \frac{\partial v}{\partial y} \tag{6}$$

$$shr = \sqrt{(\frac{\partial u}{\partial x} - \frac{\partial v}{\partial y})^2 + (\frac{\partial u}{\partial y} + \frac{\partial v}{\partial x})^2} \tag{7}$$

The total deformation rate $\epsilon$ was expressed as Eq. (8) as follows:

$$\epsilon = \sqrt{div^2 + shr^2} \tag{8}$$

The spatial scale power index β is used to describe the spatial dispersion of sea ice deformation. The smaller the value, the

more dispersed the sea ice deformation, indicating a stronger ability for deformation to propagate in space. Conversely, a larger value indicates relatively concentrated sea ice deformation. The total deformation rate $\epsilon$ calculated at different distance scales L is used to measure the spatial scale dependency characteristics of sea ice deformation (Eq. (9)) as follows:

$$\epsilon \propto L^{-\beta} \tag{9}$$



To gain a deeper understanding of the driving mechanisms behind sea-ice deformation, this study utilized daily meteorological
and sea-ice data from October 2022 to March 2023. By analysing the variations of total sea ice deformation with wind speed,
T2m, and ice thickness, a predictive model for total sea ice deformation was established to quantitatively analyse the
relationship between these factors and sea ice deformation. The model was formulated as follows:

$$f_{(x,y,z)} = \alpha_0 + \beta_1 \times e^{-(\frac{x+\beta_2}{\beta_3})^2} + \lambda_1 \times z^2 + \lambda_2 \times z^3 + \gamma \times y \times z \qquad (10)$$

Where $\alpha_0$ is the intercept term; λ, β, and γ are coefficients representing the respective influence of independent variables on
the dependent variable; and x, y, and z represent wind speed, T2m, and sea ice thickness, respectively.

## 3 Results and discussion

### 3.1 Spatiotemporal variations in atmospheric conditions

Ice motion and deformation in the Arctic region are influenced by various meteorological conditions, among which
temperature is a key factor. Notably, higher temperatures promote the melting of sea ice, whereas lower temperatures
contribute to the formation of sea ice. Wind is the driving force underlying sea ice motion, and strong winds can cause sea ice
to drift and accumulate, forming complex sea ice structures. In addition, meteorological emergencies, such as blizzards and
extreme temperature changes, can cause rapid changes in sea ice over a short period, affecting the entire Arctic sea ice
ecosystem. Overall, the complex interaction of meteorological conditions has shaped diverse sea–ice landscapes in the Arctic
region, with profound impacts on climate and ecosystems. Consequently, this study provided a brief analysis of the
meteorological conditions in the northwest Arctic region from March 2022 to March 2023.

To determine the near-surface temperature in the study area during buoy operation, we used reanalysis data provided by the
ECMWF to calculate the monthly average temperature anomalies in the study area. The results are shown in Figure 2. The
performance of temperature anomalies varied greatly during different months, with significant abnormal positive values
observed between November 2022 and March 2023 (the maximum positive temperature anomalies were 8.48, 7.06, 11.27, and
9.74 ℃). Moreover, the anomaly centres were concentrated in the central part of the study area, which undoubtedly affected
the winter sea ice conditions in the region. Conversely, the T2m air temperature in the study area was slightly higher than the
climate average from June to November 2022, with no significant abnormal centres. Quantifying sea ice motion based on
positional data from ice-based buoys is not possible because of the melting of sea ice in the summer region. Therefore, this
study focused on sea ice deformation under abnormal temperature conditions (spring, autumn, and winter) based on the buoy
data.





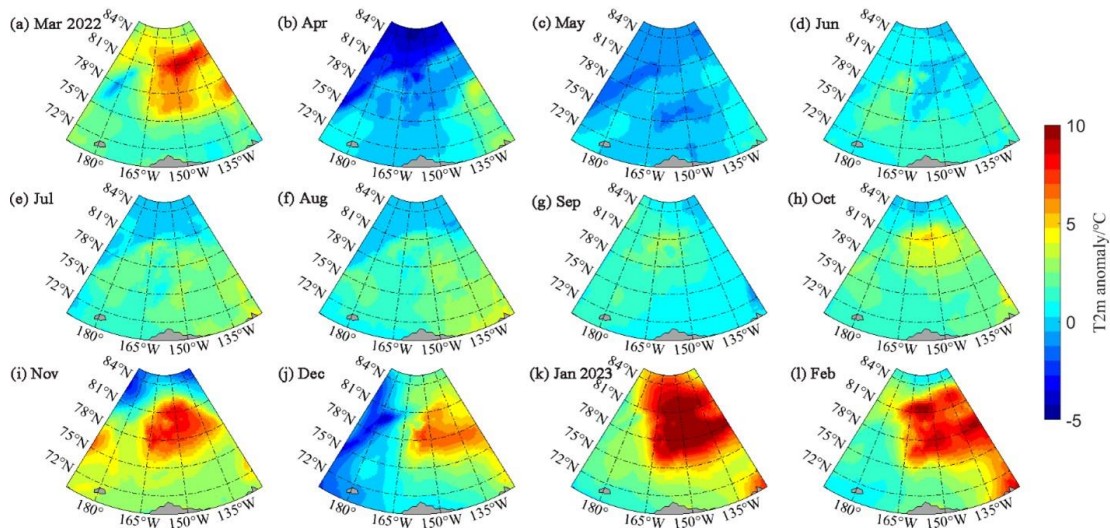

**Figure 2: Spatial distribution of monthly mean anomalies in 2-meter air temperature.**

The average sea level air pressure anomalies in different months during buoy operation in the study area are shown in Figure 3. The difference in positive and negative abnormal air pressure was significant, with the maximum abnormal positive value in January 2023 and the maximum abnormal negative value in March 2023, which was below -12 hPa (Figure 3). The anomalous phenomena of atmospheric pressure varied over time and also exhibited significant differences in different regions, with the centres of sea level pressure anomalies mostly distributed in the edge areas of the study region. This is precisely due

to the uneven spatial distribution of sea level pressure, where different pressure gradients result in differences in the distribution of the 10-m-wind fields. For example, in March 2022 and January and February 2023, the entire region of 72°–85°N and 170°– 100°E W was mainly affected by anticyclonic wind forcing; however, distinctly northerly wind fields dominated in April, June, and September 2022. Due to the wind and ocean currents, the buoy arrays shifted from their original along-latitudinal motions to radial motions (Figure 1). In addition, we calculated the time series of daily mean wind speed and T2m in the vicinity of the

ice-based buoys; the results are listed in Appendix A.





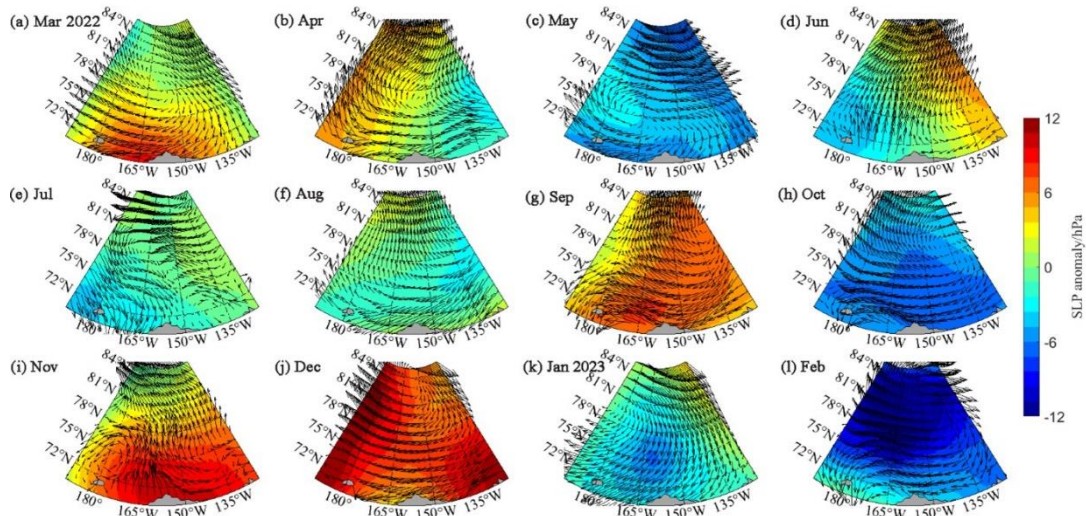

**Figure 3: Distribution of wind field and sea level pressure anomalies**

## 3.2 Sea ice deformation

### 3.2.1 Time series of sea ice deformation

We calculated the daily average total sea ice deformation using ice-based buoys with a sampling interval of 1 h. To reduce short-term fluctuations and noise and better capture the long-term trends of sea ice deformation, we calculated a moving average with a 30-day time window (Figure 4). Additionally, bilinear interpolation was used to obtain the sea ice concentration near each buoy at the corresponding times. The results are shown in Figure 4. Notably, during the summer months of 2022 (June to October), a significant decrease followed by a gradual increase in sea ice concentration was observed (Figure 4), during which the buoys were not located in ice-covered areas. Consequently, the buoy data could not be used to quantify sea ice deformation. Therefore, we defined this period (summer) as the melting season. To better compare the seasonal differences in sea ice deformation, the periods before and after the melting season were defined as spring-autumn and winter, respectively, as shown in Figure 4.

Overall, Figure 4 shows the trend of average sea ice deformation over the past year, illustrating sea ice motion. Figure 4 also shows that the total deformation in spring sharply decreased in mid-April before stabilizing at a consistent level, whereas the total deformation during the autumn-winter season demonstrated a gradual declining trend. Sea ice deformation is mainly influenced by sea ice concentration and is directly associated with sea ice thickness and wind speed. As winter approaches, the sea surface temperature decreases and sea ice thickness increases, making sea ice more resistant to fracturing and compression. Therefore, the total sea ice deformation during the autumn-winter season exhibited a decreasing trend, and the two variables were significantly negatively correlated (R = -0.68, p < 0.001).





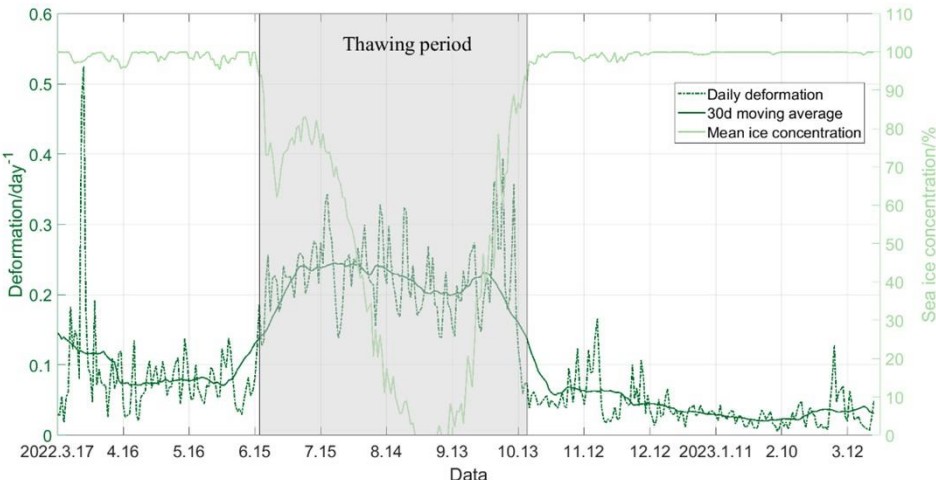

**Figure 4: Time series plot of total sea ice deformation.**

**3.2.2 Contribution of divergence and shear deformation to total deformation**

To better illustrate the seasonal variations in sea ice deformation, we calculated the monthly average shear deformation, divergence deformation, and total deformation of sea ice and analyzed the seasonal variations in the relative contributions of

each component of ice deformation to the total deformation. The trends of shear and total deformations exhibited higher consistency (Figure 5), similar to the trend changes in Figure 4. The average total deformation in spring was higher than the autumn and winter averages (0.089 and 0.042 day$^{-1}$, respectively), which may be associated with the gradual melting of sea ice in spring or the smaller spatial scale of the buoy array itself (Stern et al., 2009). Additionally, shear and total deformations decreased gradually in the autumn and winter, whereas the variation in divergence deformation was not significant across

different seasons. The relative contributions of the components of ice deformation to the total deformation exhibited significant differences, with the relative contribution of shear deformation being three percentage points higher in autumn and winter than in spring (83 and 80%, respectively) and the maximum relative contribution occurring in December (86%). Conversely, the relative contribution of divergence deformation was exactly the opposite, consistent with the findings of Lei et al.(Lei et al., 2020).

The shear deformation accounted largely for the total deformation because, in the Arctic region, sea ice is often influenced by wind and ocean currents, resulting in relative sliding and rotation between ice floes. Sliding and rotational shear deformations are two of the main components of sea ice deformation. Notably, except during the summer, the relative contribution of shear deformation tended to increase as the total deformation decreased, and the two were significantly negatively correlated (R = –0.84, p < 0.01). A possible reason for this is that when the ice layer tends to stabilize (with a decrease in total deformation),





external lateral forces (such as wind or ocean currents) are more likely to cause shear deformation of the sea ice, causing it to slide or displace relatively horizontally, which is the main manifestation of shear deformation.

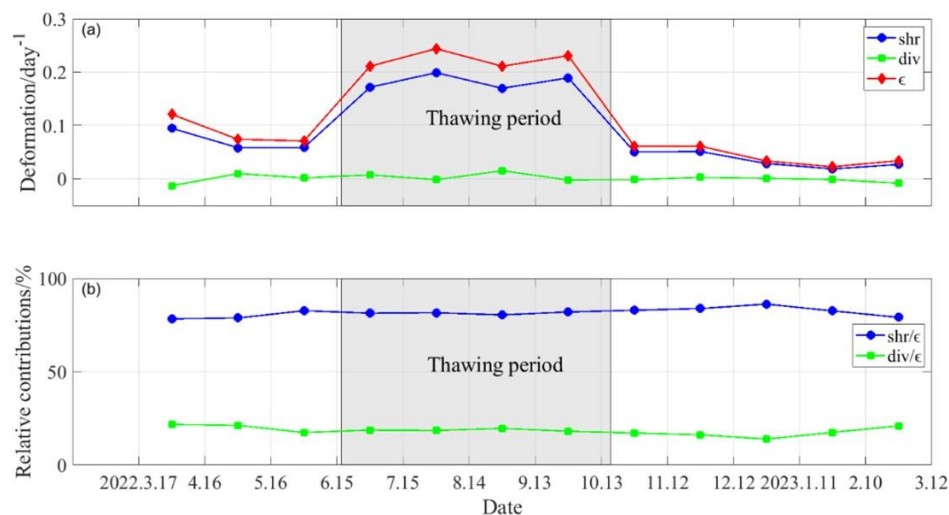

**Figure 5: Time series of monthly average sea ice deformation and the relative contributions of shear and divergence.**


### 3.3 Analysis of deformation influencing factors

When exploring global climate change, the deformation of sea ice (i.e., physical changes in sea ice) is a phenomenon worthy of in-depth study. Sea ice deformation refers to changes in the physical state of sea ice in the natural environment, including fractures, collisions, and compression. The main factors affecting sea ice deformation include changes in air temperature,
seawater temperature, sea surface currents, sea ice thickness, and atmospheric circulation patterns. Notably, the thickness of Arctic sea ice has significantly decreased over the past few decades (Kwok, 2018), which is directly associated with the rise in global average temperatures. Additionally, the recent manifestation of the 'Arctic amplification' effect (Stroeve et al., 2012) has impacted the physical state of sea ice. The decrease in sea ice not only threatens the survival of polar fauna and flora but may also lead to changes in the global climate system, such as rising sea levels and an increase in extreme weather events.
Furthermore, changes in sea ice affect the Earth's energy balance and temperature regulation mechanisms. Therefore, a thorough investigation of the factors influencing sea ice deformation is crucial for understanding and addressing global climate change. In this study, we analyzed the effects of factors such as wind, sea surface temperature, and sea ice thickness, on sea ice deformation.

### 3.3.1 Analysis of factors influencing sea ice deformation



As one of the primary driving forces of sea ice motion, the drift speed of the Arctic sea ice increases with increasing near-surface wind speeds (Yu et al., 2020). During the period 2022–2023, the wind speed did not exhibit significant seasonal variations. However, in autumn and winter, owing to the continuous freezing of sea ice, both ice speed and deformation exhibited gradually decreasing trends (Figure 6). This implies that, over the long term, the drift speed of ice is primarily controlled by seasonal-scale atmospheric forcing (Lei et al., 2020). Therefore, characterizing changes in ice speed and sea ice

deformation over longer time scales based solely on wind speed is challenging.

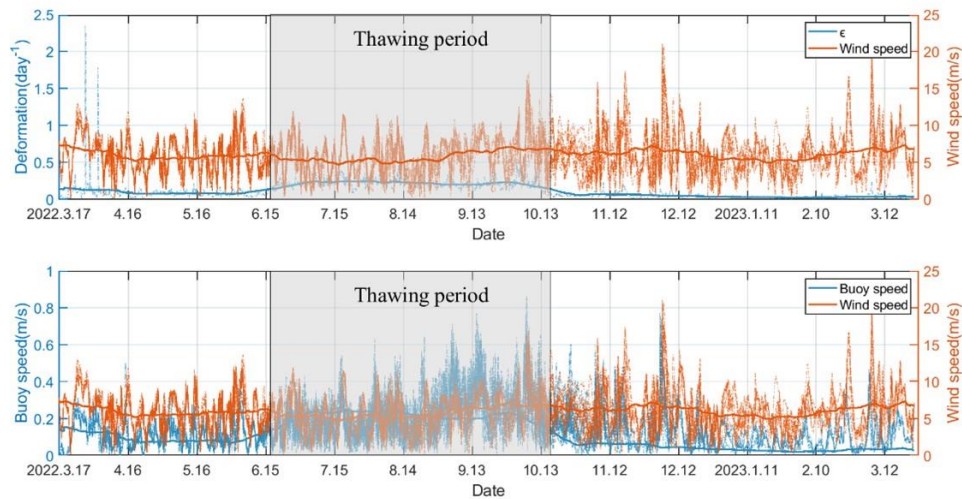

**Figure 6: Changes in wind speed and total deformation.**

Thicker multi-year ice is often less susceptible to external forcing; therefore, differences in sea ice thickness may affect sea ice movement and deformation. To reduce the impact of random errors on the results, we calculated the 30-day moving averages of wind speed, T2m, ice thickness, and total sea ice deformation and used these to calculate the coefficient of determination ($R^2$) between sea ice deformation and influencing factors to quantify their relationship. From a seasonal perspective, the impacts of wind speed on sea ice shear and total deformations were significantly higher in the different seasons

than in autumn and winter (Table 2), possibly due to the sea ice thickness being greater in the autumn and winter, making it less susceptible to external forcing and resulting in its deformation. These differences may be related to seasonal climate variations and changes in sea ice conditions, reflecting the different responses of sea ice dynamics to environmental factors in different seasons. Unlike the effect of wind speed on sea ice deformation, T2m exhibited a higher explanatory power for sea ice shear and total deformations in autumn and winter (82 and 83%, respectively). Owing to the lack of spring ice thickness

data, we only analyzed the $R^2$ between ice thickness and deformation during autumn and winter. Notably, ice thickness maintains considerable explanatory power for sea ice shear deformation and total deformation (both at 77%). The shear





deformation contributed approximately 80% to total deformation in different seasons (Figure 5), indicating that the explanatory power of different factors for both variables was generally similar across different seasons.

**Table 2: Correlation Analysis between wind speed, T2m, ice thickness, and sea ice deformation.**

| R2 | Wind speed | | | T2m | | | Ice thickness | | |
|---|---|---|---|---|---|---|---|---|---|
| | shr | div | $\epsilon$ | shr | div | $\epsilon$ | shr | div | $\epsilon$ |
| Spring | 0.69*** | n.s. | 0.64*** | 0.26*** | 0.58*** | 0.34* | - | - | - |
| Autumn and Winter | 0.21*** | n.s.** | 0.25*** | 0.82*** | n.s. | 0.83*** | 0.77*** | n.s. | 0.77*** |

Note: ***$p < 0.001$ level, **$p < 0.01$ level, *$p < 0.05$ level; n.s. indicates not significant.

### 3.3.2 Analysis of factors influencing spatial characteristics of sea ice deformation

We analyzed the impact of 2–3 factors on the spatial scale index β, including wind speed, T2m, and sea ice thickness. The β value is an indicator of the spatial heterogeneity of sea ice deformation, which is important for understanding the dynamic changes of polar sea ice. Owing to the presence of outliers in the data, we selected the median value of each factor as the threshold to consider different conditions of β. Notably, due to the limited availability of ice thickness data during spring, only the influence of meteorological elements on sea ice deformation was considered, as shown in Figure 7. When the wind speed is high (≥5.81 m/s), despite the average temperature being approximately 3 °C lower than the former, the average sea ice total deformation at the same spatial scale was higher than the latter. Furthermore, the unevenness of total sea ice deformation caused by higher wind speeds was significantly stronger (β = 0.29). In spring, when the average ice thickness was low and the air temperature was high, the internal ice stress within the sea ice decreased, making it more susceptible to external forces. Therefore, when T2m was lower, higher average wind speeds led to more concentrated sea ice deformation (Figure 7c).



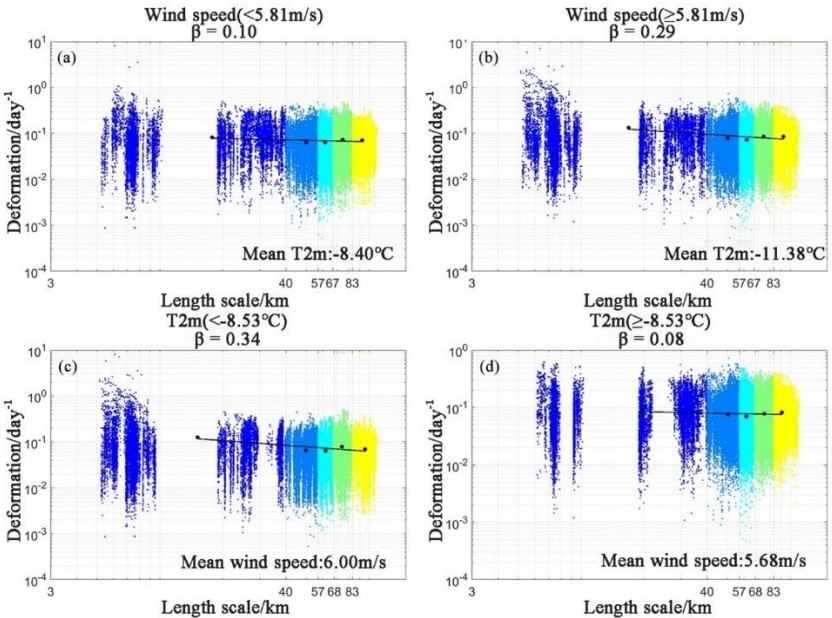

**Figure 7: Total deformation at different spatial scales under different weather conditions.**

Unlike spring, autumn, and winter, we considered the potential impact of sea ice thickness on sea ice deformation. Specifically, during autumn and winter, although significant differences were observed in wind speed, the average T2m and average sea ice thickness were comparable in both cases, resulting in a similar spatial scale index β for sea ice deformation (Figure 7a, b). This indicates that sea ice deformation is less susceptible to external forcing when the average ice thickness is larger. Therefore, the spatial characteristics of sea ice deformation in autumn and winter are more likely related to factors such as ice thickness and ocean currents. Under conditions of an average wind speed of 6.25 m/s, higher temperatures (≥ -22.59 °C), and thinner sea ice (1.11 m), the β value significantly increased from 0.18 to 0.5 compared with that in the former case. This indicates that larger T2m often decreases sea ice thickness, resulting in a significantly higher β than that in conditions with lower T2m. Therefore, under warmer and higher wind-speed conditions, the heterogeneity of sea ice deformation increases. This indicates that the increase in temperature during autumn and winter may weaken the structural strength of sea ice coupled with larger external forces (average wind speed: 6.25 m/s), increasing the unevenness of deformation. Although we used 1.35 m as the critical value for ice thickness during autumn and winter, a significant difference was observed in the average temperature between the two conditions, resulting in a smaller sea ice thickness. Coupled with similar average wind speeds under both conditions, when the ice thickness is <1.35 m, the sea ice deformation in the region exhibited greater unevenness.







**Figure 8: Analysis of the factors influencing β.**

325

Although some studies have suggested that weakening ice packs may hinder the long-distance transmission of internal stress, leading to higher β values (Stern et al., 2009), in our study, the spatial scale index β between spring and autumn-winter did not exhibit this pattern. This discrepancy is may be due to the significant difference in the spatial scales of the triangular arrays between different seasons. In spring, the spatial scale of triangular arrays was smaller (5–100 km); however, after the summer

330 sea ice melted and the movement of buoys was accelerated, the spatial scale of triangular arrays significantly increased in autumn-winter (130–400 km).



Nevertheless, sea ice deformation in different seasons showed a decreasing trend with increasing spatial scale. Overall, these results reveal the complex effects of the spatial scale, wind speed, temperature, and sea ice thickness of triangular arrays in different seasons on Arctic sea ice deformation and spatial heterogeneity. Our findings emphasize the significant impact of seasonal changes on the deformation characteristics of Arctic sea ice and reveal how environmental factors affect dynamic changes in sea ice through different mechanisms.

### 3.3.3 Establishment and evaluation of the total sea ice deformation prediction model

Wind speed, air temperature, and sea ice thickness significantly affect the total deformation and spatial heterogeneity of sea ice deformation. This study employed the least-squares method to estimate the model parameters mentioned in Sect. 2. The parameter estimation was used to minimize the total difference between the actual observed and model-predicted values. The mathematical expression for the actual model is as follows:

$$f_{(x,y,z)} = -0.03 + 10.64e^{-\left(\frac{x-1.52}{1.26}\right)^2} + 0.29z^2 - 0.17z^3 + 0.001xy \qquad (11)$$

In this study, we fitted the dataset using a multivariate nonlinear regression model to predict the complex relationship between the total ice deformation and wind speed, T2m, and ice thickness. We evaluated the performance of the regression model based on the following key indicators: residual analysis, root mean square error (RMSE), mean absolute error (MAE), $R^2$, and significance tests of the model.

The validity and reliability of the regression models were evaluated to better predict changes in sea ice deformation. First, using residual analysis, the residuals were randomly distributed around zero. Despite the sporadic outliers, the range of residuals for the prediction model was between -0.01 and 0.015 day$^{-1}$, with no significant trend change (Figure 9b), indicating that the model fitting to the dataset was appropriate. Furthermore, the RMSE and MAE between the actual values and model predictions were 0.005 and 0.004, respectively. Both metrics were relatively low, indicating a high level of prediction accuracy of the model. We also assessed the goodness of fit of the model. Notably, the $R^2$ between the actual values and model predictions was 0.87, indicating that the model explains 87% of the variance. Finally, significance tests were conducted for each coefficient of the model. The significance of the coefficients was evaluated using *t*-tests, and the overall significance of the model was assessed using the *F*-test. The results showed that the coefficients of all independent variables were statistically significant at the level of $p < 0.01$, and the *F*-test of the overall model was also significant ($p < 0.01$), demonstrating the effectiveness of the model. Notably, the *t*-test results for each predictor variable were significant in predicting the total ice deformation.





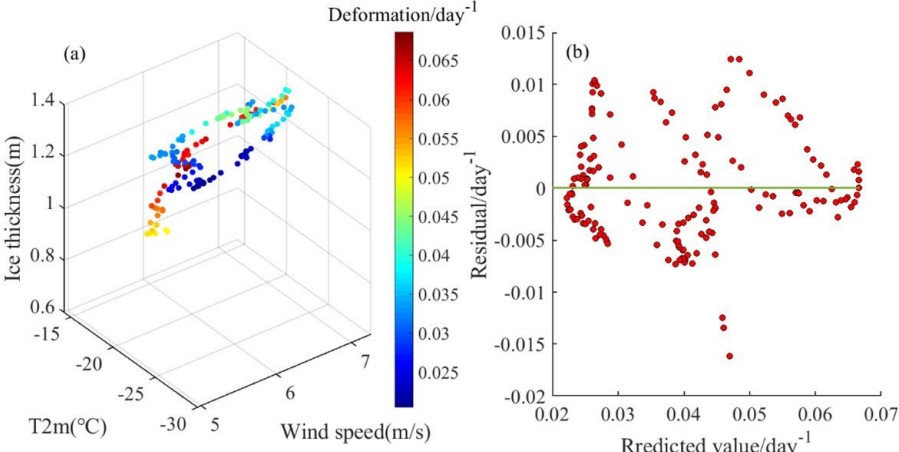


**Figure 9: Regression analysis results and residual distribution plot.**

The regression analysis results (Figure 9a) indicate that wind speed, T2m, and ice thickness exert significant impacts on ice deformation. Specifically, the increase in wind speed was positively correlated with ice deformation, suggesting that stronger

winds led to more pronounced ice movement and fracturing. The increase in temperature was positively correlated with ice deformation, reflecting the melting and structural damage to sea ice caused by the temperature rise. However, the increase in sea ice thickness was negatively correlated with ice deformation, indicating that thicker sea ice is more stable and less sensitive to environmental changes. These findings reveal that wind speed, temperature, and sea ice thickness are the key factors influencing Arctic sea ice deformation. Furthermore, these results are important for predicting future sea ice changes,

formulating corresponding environmental protection policies, and providing guidance for maritime safety in the Arctic region. Future research should explore the more complex interactions between these factors and sea ice deformation, as well as the potential impacts of global climate change on this relationship.

## 4 Conclusions

This study explored the spatiotemporal variations in atmospheric conditions in the study area and simultaneously analyzed the

changes in sea ice deformation in the western Arctic from March 2022 to March 2023 using data from ice-based buoys. We discussed the impacts of different factors on sea ice deformation, with particular emphasis on the factors influencing the spatial characteristics of ice deformation. Additionally, we established a predictive model based on three factors (wind speed, T2m, and ice thickness) to predict total ice deformation.

Our main results are as follows:

First, under the influence of geostrophic winds, the buoy array changed from its original zonal flow to a meridional flow. After the melting season, the average spatial scale of the buoy array significantly increased from the original 5–100 to 130–400 km, providing an opportunity to discuss the factors influencing the spatial characteristics of sea ice deformation at different spatial



scales. The total ice deformation gradually decreased in different seasons. Additionally, in the autumn and winter, the average total sea ice deformation was lower than in spring (with average total ice deformations of 0.09 and 0.04 day$^{-1}$ in spring and

autumn-winter, respectively) because of the continuous solidification of sea ice. In addition, shear deformation accounted for approximately 80% of the total deformation in different seasons, highlighting why different influencing factors have similar explanatory powers for shear and total deformation in different seasons.

Second, in the analysis of the factors influencing the spatial characteristics of ice deformation, the total ice deformation tended to gradually decrease with increasing spatial scale. During spring, higher wind speeds or higher T2m values often result in

higher β values. This suggests that when wind speeds are higher or temperatures are warmer, ice deformation tends to be more concentrated. However, when wind speeds are lower or temperatures are cooler, deformation tends to be more dispersed, indicating a greater ability for deformation propagation. However, because of the smaller average ice thickness during spring, the influence of external forcing on the spatial characteristics of ice deformation was greater than that of the ice thickness itself. Conversely, deformation sensitivity to external forces decreased in the autumn and winter owing to the gradual increase in

internal forces within the sea ice. In addition, the spatial scale index β exhibited no significant changes (0.25 and 0.24) under different wind speed conditions. In contrast to the effect of wind speed on the spatial characteristics of ice deformation, the impact of different T2m and ice thickness on β significantly varied. When the temperature was low and the average ice thickness was high, sea ice deformation occurred more uniformly. In summary, during spring, external forcing was the main factor influencing the spatial characteristics of ice deformation due to the relatively small average ice thickness. However, in

autumn and winter, due to the larger average sea ice thickness, external forcing becomes less significant, and the T2m/ice thickness becomes the primary factor affecting the spatial characteristics of sea ice deformation.

Finally, we established a multivariate nonlinear regression model to examine the influences of wind speed, T2m, and ice thickness on total ice deformation. Notably, the overall model and coefficients passed the significance tests. The residuals of the model mainly fall within the range of -0.01–0.015 day$^{-1}$, with the RMSE and MAE between the true and predicted values

of total ice deformation being 0.005 and 0.004, respectively, both at a relatively low level. Additionally, the $R^2$ value between the true and predicted values of the total deformation was 0.87, indicating a good fit for the predictive model. In summary, these evaluation metrics collectively indicated that our regression model was effective and reliable for predicting total ice deformation. The above-described analysis demonstrated that when sea ice thickness remains relatively constant, total ice deformation increases gradually with increasing wind speed; however, when external forcing factors exhibit minimal variation,

total ice deformation increases gradually with increasing T2m or decreasing average ice thickness.

The Arctic sea ice is a key indicator of climate change. Studying the factors influencing total ice deformation in the Arctic provides deeper insights into the impact of climate change on sea ice and data to support climate change awareness. This study contributes to improving the predictive capabilities of future Arctic sea ice deformation, offering a scientific foundation for climate change mitigation and adaptation efforts. However, although the model performed well in this study, potential

multicollinearity issues may still exist. The influence of atmospheric factors on the characteristics of sea ice motion is a complex process; therefore, further exploration of other possible influencing factors on total ice deformation remains warranted.



## Appendix A

We resampled the 18 sets of buoy data, extracted the daily mean wind speed and T2m at the buoy locations, and calculated the time series variation of each data. The results are shown in Figue A1.


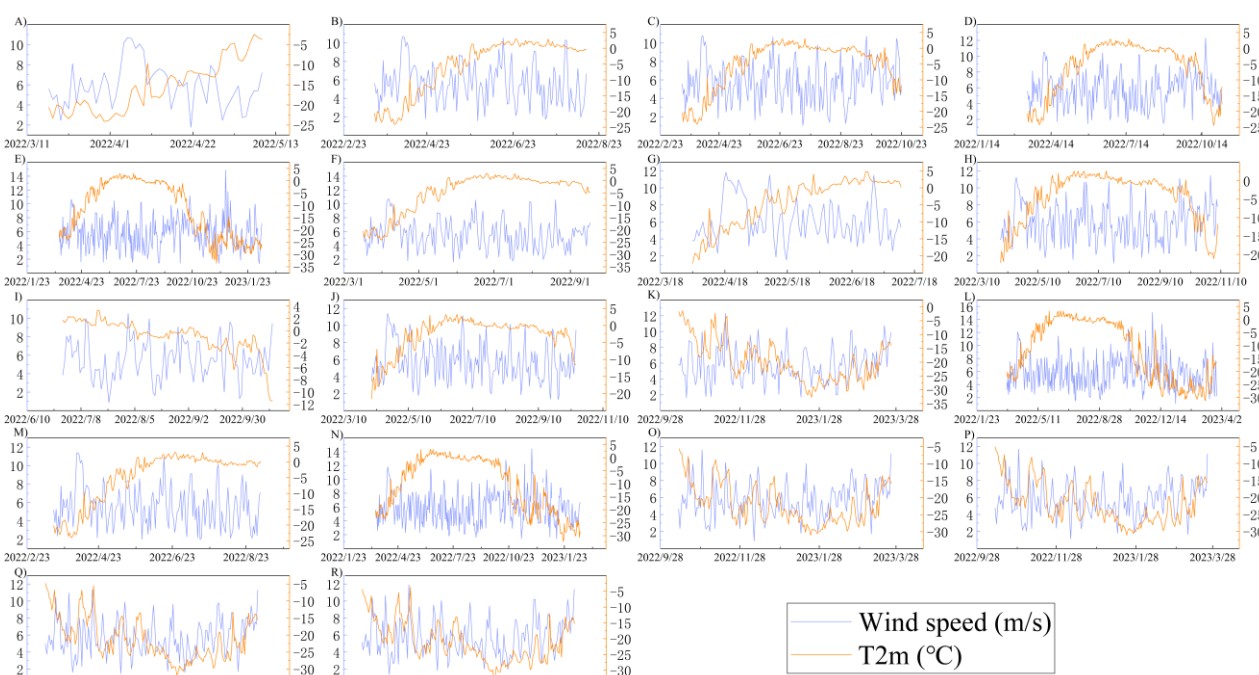

**Figue A1: Time series variation of daily mean wind speed and T2m from 18 groups of buoy networks.**

## Data availability

Data will be made available on request.

## Author contribution

**Tongliang Yan:** Writing – original draft, Conceptualization. **Lei Ji:** Investigation, Formal analysis, Data curation. **Bowen Zhu:** Writing – review & editing, Validation. **Guangyu Zuo:** Methodology, Validation, Resources. **Xiaomin Chang:** Resources, Conceptualization. **Yinke Dou:** Project administration, Funding acquisition.




## Competing interests

The authors declare that they have no conflict of interest.

## Disclaimer

Publisher's note: Copernicus Publications remains neutral with regard to jurisdictional claims made in the text, published maps, institutional affiliations, or any other geographical representation in this paper. While Copernicus Publications makes every effort to include appropriate place names, the final responsibility lies with the authors.

## Acknowledgements

The authors thank the National Key R&D Program of China (No. 2022YFC2807601), the Shanxi Provincial Natural Science Foundation for the General Program (No. 20210302124054), and the Shanxi Basic Research Program (No. 20210302124318) for supporting this study. We also thank all the scientists for their valuable research in this study.

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
