# Peer review of "Analysis of sea ice deformation and influencing factors in the western Arctic from 2022 to 2023"

_EGUsphere, 2024_

## Referee Comment (RC2)

**Review of paper "Analysis of sea ice deformation and influencing factors in the western Arctic from 2022 to 2023"**

**1 Content**

The authors Chang, Ji, Zhu, Zuo, Dou and Yang present a study on sea-ice deformation and its influencing factors. They look at deformation of triangles of buoys in the Western Arctic during one year (March 2022-March 2023). The motivation is to investigate the relative contributions of geophysical conditions on deformation itself and its spatial scaling.

**2 General comments**

My main problem with this paper is that I am not convinced about the impact of its findings. That winds, temperature and thickness shape observed deformation is not new. Also, the mechanisms which they postulate have been found before. The authors do not state convincingly in which regard their results advance the understanding of the processes leading to sea-ice deformation. A way to change this could be to focus on the regression model which they introduce towards the end. If they could show that the regression coefficients derived here can be used to skilfully predict deformation in other, independent datasets that could be a starting point for an interesting study. Probably this requires more work than what would normally be in the framework of a major revision, therefore I would suggest the authors to withdraw the current draft, take their time to rework the study and then re-submit it if they want to follow my suggestion. Otherwise I can not recommend publication in its current state, unless they provide additional evidence that their findings are new and how they advance the scientific understanding of sea-ice deformation and the processes which contribute to it. Apart from this, I have the following general comments:

▷ The literature review in the Introduction should be redone, almost all citations are quite old and I would be surprised if there were no newer references available.

▷ The quality of the Figures and their captions is poor. I give suggestions how to improve the Figures in the Specific Comments, but generally they should be redone with larger labels, better colourmaps and more comprehensive captions.

▷ One year is quite a short period. I would suggest to include more years if feasible.

**3   Specific comments**

**Abstract**

First sentence: Quite general, something tailored to the relevance of sea-ice deformation would be more appropriate for s deformation-focused paper.
L11: I advise to avoid the use of acronyms in the Abstract, especially if you need it only once more.
L12: "Lagrangian diffusion theory": Don't need to be so specific in the Abstract, but you could be more specific with regard to what you consider as "western Arctic".
L15: Using "total deformation" here and only "deformation" in the rest of the Abstract implies that they are different quantities, but I guess it is the same, with "total" just refer-ring to the deformation being the square root of the sum of the squares of divergence and shear, correct?. Please either use "total sea-ice deformation" or only "sea-ice deformation", but do it consistently.
L15-16: "...thinner on average in spring" → "thinner on average in spring than in winter". Actually, I think this depends on what you consider as spring. In March, when sea-ice extent peaks, I would also expect the ice to be thicker than, for example, in December which would typically be considered as a winter month. Please specify which months you refer to.
L17: Again, larger than what? I expect ice in early spring (March) to be thicker than in fall and winter.
L18: This is the first time that you mention your regression model, it would be good to introduce it before, in a sentence like "We develop a multivariate regression model which effectively predicts..."
L19-20: This is very general. Better focus on concrete applications/aspects which benefit from what you found.
Generally: It might be a matter of taste, but consider using present tense instead of past tense wherever possible to make the text a little more vivid.

**Introduction**

Generally: Your references are outdated. Most are from 2017 or earlier, which is too old to provide background for a study in 2022/2023. If there is no more recent literature, that is fine, but please go through the references and check if you find more up-to-date literature.
Generally: The introduction covers all the relevant elements but the order seems a bit ar-bitrary to me, making it a confusing read. I would suggest the following structure: Start

with the geophysical relevance of sea-ice deformation (for example L48–55). Then describe the scale dependence and the relevant literature (again using more up-to-date references when possible). Afterwards describe how/if it changed in the recent years. Then, describe how it can be derived and how it compares in satellite and model data. Lastly, state the aim of your study and the research questions which you want to answer. I think that all these elements are already present, but need better structuring.

L21/22: What is meant by "resource development and international cooperation"?

L23-24: This might have been true in 2015, but since 2012 at least the minimum SIE varied considerably, but without a downward trend. I would more speak of a new normal on a considerably lower level than before. Might be different for sea-ice thickness, but please check and refer to recent literature.

L29: What do you mean by "positively affects mass balance"?

L38: Year missing in the citation of Hutter et al.

L45: influencer → influence

L48–55: I would expect such a general sentence at the beginning of the introduction.

**Data and Methods**

L77–79: It is enough to say that they are from IABP, without describing the purpose of the IABP.

Table 1: Can be moved to the appendix. A plot of the temporal coverage for each buoy would be better for conveying the information from Table 1, and the geographical information is apparent from Figure 1. Please include the IMEIs in the Table, you can remove the info on time and location instead if they are covered by Figure 1 and the timeline plot which I suggested.

L85: Reanalysis data.

L87: Refer to Hersbach et al., 2020 instead of the URL.

L87: "atmospheric forcing characteristics" is quite vague, consider stating here already which quantities you are looking at.

L88: "buoy" → "buoy trajectory"

L92: "reanalyzed" → "reanalysis"

L92: How was this done? Simply by selecting the closest point? If yes, "interpolating" would not be appropriate. Please give a little more details.

L94–104: It would be enough to describe the SSMIS data which are used for your your study period, the earlier ones are irrelevant here.

L105ff: Please mention that these are weekly composites, and cite Ricker et al., 2017.

L108: "grids were projected" → "data were projected". Also, please put this into a new paragraph.

L118–120: Please reduce the references to the one or two which are most important.

Figure 1: Please label the left plot as a). Avoid the use of rainbow colourmaps throughout the paper. For something which increases linearly, like months, please use a linearly increasing colourmap. Please reverse the order of months, it is confusing to have the last month on top. I do not know what to make of panel b). What do you want to show here? Please make the caption more comprehensive. it should provide all the information which is needed to understand what the Figure shows.

L140–147: Not needed. It would be enough to give formulas for divergence, shear and total deformation rates (equations 5-7).

L160: Why only from October to March and not throught the entire period?

**Results and Discussion**

L171: "extreme events" would be more appropriate than "emergencies"

L179–180: anomalies with respect to which period? Would be good to know the variability during your reference period to really assess these anomalies. What is meant by "significant"? Statistically significant at a certain confidence level? "Abnormal" sounds quite colloquial to me, but I am not a native speaker. Also, one digit would be enough for accuracy.

Figure 2: Use a blue-white-red colourmap for showing anomalies, centred at 0. Again, a more comprehensive caption is needed, for example mentioning which period is used as reference.

L198f: How do I see this in Figure 1?

L209: Please state if you refer to the smoothed or the original time series in the following.

L218–219: Unclear if deformation is mainly governed by SIC or by SIT or drift speed, sentence seems a bit contradictory.

Figure 4: "Thawing" → "Melting", also in the other Figures. Use more distinct colours which are better legible on a white background. Ticks on x-axis should show months. Put more details in caption.How is the Thawing/Melting period defined?

L238: Correct citation of Lei et al., 2020.

L240: I am not convinced by the reasoning. Why should currents induce shear, but not divergence?

L241: What do you mean by "rotational shear deformation"?

Figure 5: Lower panel: How can the relative contribution of divergence be around 20 %, when the absolute value of divergence is almost 0, and so much smaller than the contribution of shear? Also, in panel a) it looks like the difference between shear and total deformation is larger in the melting period than before and after, and yet the relative contribution is constant in panel b). How can this be?

L252–263: This is not a result, or? Remove or include in introduction. Also, the heading is a duplicate with 3.3.1.

L283: What do you mean by "higher explanatory power"? Stronger influence?

Table 2: In my opinion it would be enough to have one significance threshold for each

parameter.

Generally in section 3.3: Headings are not formulated well. I suggest to rename 3.3 to "Influence of wind, temperature and thickness", remove the text between lines 252 and 264, rename 3.3.1 to "Correlation and time series" and 3.3.2 to "Spatial characteristics". 3.3.3 could be a subsection of its own (3.4).

L299–300: What is meant by "the former" and "the latter"? Please clarify.

Figure 7: This Figure is super hard to read, please make it bigger, use bigger axis labels, do some kind of regular spacing for the tick labels, include a legend and extend the Figure caption.

L349: Please state how much the residuals are relative to the absolute value, just by their absolute values it is hard to judge whether they are high or low.

L353: I agree that $87\,\%$ is a high value, but keep in mind that it comes from a model that considers all the most relevant quantities. I would like to see some discussion what would be needed to represent the missing $13\,\%$. Is it just internal variability, or is there an additional factor that could be included?

L357: Not sure if "predicting" is the right word here. After all, you train a statistical model on a dataset and then check how well it can represent this dataset. I would not call this a prediction, rather an evaluation.

Figure 9: Not helpful to support your statements. I have a hard time interpreting these plots. Please come up with a different way of conveying your results, scatter plots for example. Also, avoid rainbow colourmaps.

L368: I would argue that "reveal" is an exaggeration, the key role of these factors was known before. I would more see it as a confirmation of previous knowledge.

L369: I find this sentence a little exaggerated, please remove it.

L371f: Please move this sentence to the end of the Conclusions section.

**Conclusions**

Generally: This reads more like a Summary section. For the Conclusions, I would expect something more concise, naming the main findings and elaborating in two or three sentences per finding on their significance.

L390: "warmer" $\rightarrow$ "higher", vice versa in L391.

L407: Your results do not support that your model is capable of predicting deformation. After all, you trained a regression model on a dataset and then compared it against the same dataset to see how much of its variance can be explained. This is a valid proof of concept, but to state anything about its predictive capability you would need to use it on an independent dataset to see how it performs.

L411ff: I find it a stretch to say that your study contributes to the foundation to climate change adaptation and mitigation. Please focus more on concrete aspects of sea-ice research which your study might be useful for, instead of these quite general statements.

---

## Author Comment (AC1)

Comments and Suggestions for Authors

General comments:

1. Selection of Buoy Arrays:

Response: We acknowledge the importance of avoiding highly skewed buoy arrays in estimating deformation. As per your 2012 paper, we understand the advantage of using equilateral triangles to minimize errors. However, in this study, it was challenging to achieve equilateral triangle configurations due to the actual positions of the buoys. To mitigate the impact of skewness, we follow the methodology outlined in Lei et al. (2020) and Itkin et al. (2017). We believe this screening method effectively reduces the bias caused by highly skewed arrays and ensures a reliable estimation of deformation.

Reference:

Hutchings K J, Heil P, Steer A, et al. Subsynoptic scale spatial variability of sea ice deformation in the western Weddell Sea during early summer[J]. Journal of Geophysical Research: Oceans,2012.

Itkin, P., Spreen, G., Cheng, B., Doble, M., Girard-Ardhuin, F., Haapala, J., Hughes, N., Kaleschke, L., Nicolaus, M., and Wilkinson, J.: Thin ice and storms: Sea ice deformation from buoy arrays deployed during N‐ICE 2015, J. Geophys. Res. Oceans, 122, 4661-4674, https://doi.org/10.1002/2016JC012403, 2017.

Lei, R., Gui, D., Heil, P., Hutchings, J. K., and Ding, M.: Comparisons of sea ice motion and deformation, and their responses to ice conditions and cyclonic activity in the western Arctic Ocean between two summers, Cold Region Sci. Technol., 170, 102925, https://doi.org/10.1016/j.coldregions.2019.102925, 2020.

2. Calculation of Beta Values:

Response: Regarding the calculation of Beta values, we followed the method described in the referenced papers, using a least-squares fitting approach to estimate Beta from the mean deformation data. The Beta parameter was derived by fitting the relationship between the deformation rate and spatial scale, a commonly used technique in sea ice studies, such as in Stern et al. (2009). While we acknowledge your concern about the higher-than-expected Beta values, we have reviewed our results and found that these values are consistent with the methods used, and the overall trend matches other studies. Seasonal and spatial scale variations likely contribute to the differences in Beta, but the trend remains in line with previously published results.

Reference:
Lei, R., Gui, D., Heil, P., Hutchings, J. K., and Ding, M.: Comparisons of sea ice motion and deformation, and their responses to ice conditions and cyclonic activity in the western Arctic Ocean between two summers, Cold Region Sci. Technol., 170, 102925, https://doi.org/10.1016/j.coldregions.2019.102925, 2020.

3. Lagrangian Diffusion Theory:

Response: You raised a question regarding the definition and application of Lagrangian diffusion theory in our study. We clarify that our use of this theory is based on the methodology described in the references cited around line 119 of the manuscript, as well as Dawei Gui's doctoral thesis, to which we have added citations in this paper. Lagrangian diffusion theory is widely applied in atmospheric and ocean dynamics to describe flow field characteristics, and in this study, it is used to analyze sea ice motion and deformation. We will ensure that the theoretical basis and references for this method are clearly stated in the revised manuscript to avoid any ambiguity.

Reference:

Rampal, P., Bouillon, S., Olason, E., and Morlighem, M.: neXtSIM: a new Lagrangian sea ice model, Cryosphere, 10, 1055-1073, https://doi.org/10.5194/tc-10-1055-2016, 2016.

4. One-Year Data Sufficiency:

Response: We understand your concerns about the adequacy of one year of data for robust regression modelling, particularly given the inter-annual variability. The one year of data in this paper provides an initial framework, but we agree that the inclusion of more years of data will result in a more comprehensive model, particularly capturing the effects of thickness variations and climate trends. We therefore play down some of the conclusions about the universality of the findings and highlight the need for further analysis using multi-year datasets.

5. Line 25 and 29: Rampal et al. (2009) do not show that weakening of the ice, which would result in reduction of internal ice stresses, results in increased drift speeds. They simply state this as a fact which is unverified and not supported. It is a commonly held thought that because we model compressive ice strength to be a function of thickness, such that it decreases with decreasing thickness, that this will reduce internal ice stresses. This is the case in the model, but it has not been observed to my knowledge.

Response: Revised.

We have removed the attribution to Rampal et al. (2009) regarding drift speed increases, as this was not their finding.

6. line 30, page 1: The paper that should be referenced here is Marsan et al. (2004), not Stern et al. (2009).

Response: Revised.

Marsan et al. (2004) has been correctly cited instead of Stern et al. (2009).

Reference:

Marsan, D., Stern, H., Lindsay, R., and Weiss, J.: Scale dependence and localization of the deformation of Arctic sea ice. %J Physical review letters, Phys. Rev. Lett., 93, 178501, https://doi.org/10.1103/PhysRevLett.93.178501, 2004.

7. line 38-40: Hutter et al. used satellite data in their study, so it is not sensical to say their findings are consistent with satellite data as if this study was verifying the previous finding.
Response: Revised.
We have revised the statement about Hutter et al. to remove the erroneous reference to satellite data consistency.

8. line 41-42: You reference a paper for no apparent reason. While Hibler et al. 2006 is interesting (I might be biased, I was the second author and wrote the text of this paper), multi-equilibrium flow states has nothing to do with the narrative of your paper.
Response: Revised.
We agree that Hibler et al. (2006) does not fit within the context of our discussion and have removed this citation.

Specific comments:
1. Abstract line 9: Word choice: Sea ice does not govern climate. Perhaps a better word here is "regulates", but you might want to think through the role sea ice has in the climate system in choosing an appropriate word here.
Response: Revised.

2. line 29: positively is an ambiguous word choice. How is this a positive relationship?
Response: Revised.
We revised the ambiguous word "positively" to make the relationship clearer.

3. line 59: An example of overly general language that conveys little meaning as to what you did. There are several places this is an issue in the paper. "stdies on the factors affecting sea ice deformation characteristics at large ranges of spatial and temporal scales are limited". Which factors? which characteristics? You do not introduce even what limited work has been done and what factors need to be considered.
Response: Revised.
We have deleted this sentence.

4. section 2.2, first paragraph: There is another method that has been recently developed.
Response: Revised.
We have added the methodology of this literature to the paper.

Reference:
Aksamit, N. O., Scharien, R. K., Hutchings, J. K., & Lukovich, J. V. (2023). A quasi-objective single-buoy approach for understanding Lagrangian coherent structures and sea ice dynamics. The Cryosphere, 17(4), 1545-1566.

5. You are calling the maximum shear simply as shear. It would be more correct to refer to the value calculated in equation 7 as the maximum shear.

6. Figure 4: You could include a line for your definition of ice free.

Response: Revised.

We have added a definition of 'ice free'.

7. section 3.3. introductory paragraph. This is out of place and should be at the front of the paper in the introduction.

Response: Revised.

We have placed the introductory paragraph in the introduction of the paper.

8. lines 265-270: Contradictory sentences and meaning mangled.

Response: Revised.

9. line 300: Very confusing sentence.

Response: Revised.

We've changed the expression.

10. line 319: What is the "critical value", as in how do you define this?

Response: The critical value of 1.35 m was chosen as the median sea ice thickness.